# Avian MHC Evolution in the Era of Genomics: Phase 1.0

**DOI:** 10.3390/cells8101152

**Published:** 2019-09-26

**Authors:** Emily A. O’Connor, Helena Westerdahl, Reto Burri, Scott V. Edwards

**Affiliations:** 1Department of Biology, Lund University, SE-223 62 Lund, Sweden; emily.o_connor@biol.lu.se (E.A.O.); helena.westerdahl@biol.lu.se (H.W.); 2Department of Population Ecology, Institute of Ecology & Evolution, Friedrich Schiller University Jena, 07737 Jena, Germany; burri@wildlight.ch; 3Department of Organismic and Evolutionary Biology and Museum of Comparative Zoology, Harvard University, Cambridge, MA 02138, USA

**Keywords:** MHC genes, birds, disease resistance, orthology, life history, gene duplication, long-read sequencing, high-throughput sequencing, concerted evolution, ecology

## Abstract

Birds are a wonderfully diverse and accessible clade with an exceptional range of ecologies and behaviors, making the study of the avian major histocompatibility complex (MHC) of great interest. In the last 20 years, particularly with the advent of high-throughput sequencing, the avian MHC has been explored in great depth in several dimensions: its ability to explain ecological patterns in nature, such as mating preferences; its correlation with parasite resistance; and its structural evolution across the avian tree of life. Here, we review the latest pulse of avian MHC studies spurred by high-throughput sequencing. Despite high-throughput approaches to MHC studies, substantial areas remain in need of improvement with regard to our understanding of MHC structure, diversity, and evolution. Recent studies of the avian MHC have nonetheless revealed intriguing connections between MHC structure and life history traits, and highlight the advantages of long-term ecological studies for understanding the patterns of MHC variation in the wild. Given the exceptional diversity of birds, their accessibility, and the ease of sequencing their genomes, studies of avian MHC promise to improve our understanding of the many dimensions and consequences of MHC variation in nature. However, significant improvements in assembling complete MHC regions with long-read sequencing will be required for truly transformative studies.

## 1. Introduction

Birds and the major histocompatibility complex (MHC) have a long and special relationship. The domestic chicken (*Gallus gallus domesticus*) was among the first species to have its MHC characterized at the functional level, and the chicken MHC has been the major non-mammalian vertebrate model for MHC structure and genomic organization [1,2,3,4,5]. The chicken MHC has also yielded some of the clearest associations between disease resistance and genotype, a link speculated to be associated with the relatively simple and compact structure of the chicken MHC when compared with that of mammals [1,5,6]. However, despite decades of immunological work on the chicken, interest in the MHC among ornithologists with a focus on ecology and evolution came not from studies on chickens but from mammals. The possibility that MHC variation might influence mating preferences as well as disease resistance were revealed for the first time outside of laboratory strains by the landmark studies of Wayne Potts [7,8]. These studies in semi-natural populations of mice, alongside a growing interest in the role of MHC in mate choice and kin recognition [9], catalyzed the first explorations of MHC variation in natural populations of birds [10,11,12,13,14,15]. The long-term goal of these avian MHC studies was to find the genes influencing fitness and behavior in the wild [11,12,13,14,15], and to aid the conservation of biodiversity through the analysis of genes with important functions in conservation biology [16]. Currently, there is a strong interest among evolutionary biologists in genes and regulatory regions underlying phenotypic variation, with many recent exciting examples from birds [17,18,19]. Genes of the MHC were the first candidates for variation in fitness and fitness-related phenotypes in birds, including disease resistance, plumage brightness, as well as mating preferences and success [11,13,14]. The extraordinarily rich body of theory relating to signals of superior disease resistance, sexual selection, mating patterns, and behavior [20,21,22] was inspired and compellingly exemplified to a large extent by birds; this body of theory also provided a fertile foundation for the flood of avian MHC studies beginning in the latter half of the 1990s [13,14,23,24].

This review provides an update of studies of MHC evolution in birds, including recent advances in our understanding of structural evolution of the avian MHC, as well as new insights into aspects of avian ecology and evolution provided by the MHC. Our update is timely because the field of avian MHC is poised for advances in our understanding of the structural genetic variation of MHC due to the adoption of affordable and accessible long-read sequencing technology: phase 1.0 of the genomic era. Our focus is on species that are models in ecology and evolutionary studies, such as songbirds (oscine passerine birds), and less so on avian species that are immunological models, such as the chicken, other gamebirds (Galliformes), and waterfowl, which have been reviewed elsewhere [2,3,4]. An assumption of our overall perspective is that little can be learned about the evolution of MHC genes in birds without comparisons among species. The publication of genome sequences for a large number of bird species has opened the door to comparative analyses of genomic regions of interest [25]. However, the incomplete assembly of the MHC region in genomes generated using short-read sequencing technology has limited the scope for whole genome data to be used in comparative analyses of avian MHC.

Recent phylogenetic studies of MHC evolution using data from targeted MHC genotyping studies, which we review here, have revealed exciting long-term patterns of gene duplication and divergence. The quality of non-chicken avian genomes lags behind that of the chicken. Nevertheless, we can start to see some clear trends in the evolution of MHC structure and gene number across the avian tree of life [3]. Even from early rudimentary studies using genomic cloning [23,24,26,27,28,29,30,31], it was evident that the paradigm put forward by the chicken MHC, with less than 20 genes and spanning less than 100 kb, did not represent birds generally; indeed, the zebra finch MHC genomic region(s) exceeds 700 kb [32,33]. The emerging picture of MHC evolution in birds is that species within the Galliformes appear to have smaller MHC genomic regions than most other species of birds [5,28,32,34,35,36,37,38] (but Shiina et al. 2004 provides evidence of more extensively duplicated MHC genes in the Japanese quail (*Coturnix japonica*) [39]). In this respect, the chicken, with its ‘minimal essential MHC’, may be the ‘odd duck’ when it comes to their genomic organization of the MHC. Moreover, expression of MHC genes in non-galliform birds can be highly complex, with multiple class I and II genes expressed in some songbirds [23,24,40,41,42], suggesting that the chicken may also be an outlier with regard to MHC expression (but see Drews et al. 2017 [43]). There is conclusive evidence for classical and non-classical MHC class I (MHC-I) or MHC class IIB (MHC-IIB) in Galliform species such as the chicken, the turkey (*Meleagris gallopavo*), the black grouse (*Tetrao tetrix*), and the golden pheasant (*Chrysolophus pictus)* [5,36,44,45,46,47,48,49]. Possessing both classical and non-classical MHC genes appears to be a taxonomically widespread phenomenon because it has been observed in primates, fish, amphibians, and reptiles [50,51,52,53,54]. Classical MHC genes are usually highly expressed, polymorphic, and have a well-established function in presenting antigens to T-cells [55]. On the other hand, non-classical MHC genes usually exhibit lower polymorphism, are only weakly expressed, and may have functions beyond classic antigen presentation [56,57,58]. The detailed studies required to confirm the presence of non-classical MHC genes outside of Galliformes are currently lacking, but evidence from allelic polymorphism and expression patterns suggests that a number of species within the Charadriiformes, Pelecaniformes, and Passeriformes may also possess both classical and non-classical MHC-I and MHC-II genes [42,43,59,60]. In songbirds, these putatively non-classical MHC-I genes are not orthologous to MHC-Y, a second polymorphic MHC region also on chicken chr 16 [4], making it likely that non-classical MHC genes have arisen on multiple independent occasions across the evolution of birds [43]. Furthermore, the non-classical MHC genes described in mammals, fish, and birds all appear to have independent origins [51,61]. We use the terms MHC-I and MHC-II throughout this review to refer only to classical MHC genes unless otherwise stated.

We also discuss recent efforts to integrate studies of MHC variation in long-term ecological studies, for which birds provide many excellent examples. Studies on wild birds reveal how MHC evolves in nature and impacts fitness. The detailed mechanistic understanding of MHC, generated by studies on chickens, help us to interpret the biological relevance of the patterns of MHC variation we see in nature. In the last ten years, ornithologists and evolutionary biologists have gone to great lengths to understand the evolutionary forces driving MHC polymorphism. Here, we hope to provide an overview of this quiet revolution, which we believe will foster a second renaissance of MHC studies in birds.

## 2. The Avian Major Histocompatibility Complex (MHC) Enters the Genomic Era

### 2.1. Technical Advances: Large-Scale MHC Structure

Twenty years have passed since the first genomic structure of an avian MHC, that of the chicken, was characterized [5]. However, despite tremendous technological advances, strategies for the genomic characterization of bird MHCs have barely changed. Most of the recent genomic characterizations of bird MHCs, such as those of the black grouse (*Lyrurus tetrix*), crested ibis (*Nipponia nippon*), and oriental stork (*Ciconia boyciana*), have employed sequencing of MHC-containing fosmids or BAC clones as the preferred strategy [35,36,62,63]. Only in a very few instances have MHCs been characterized from *de novo* genome assemblies; even in these instances MHC structures were supported by sequencing of BAC clones [32]. Today, the cost of BAC-library construction, screening for MHC-containing clones, and subsequent sequencing likely exceed the costs of a typical *de novo* assembly of a bird genome. This begs the question why research on bird MHCs has made only limited use of existing genome assemblies [25]. Thus far, high-throughput sequencing and traditional genome assembly approaches have not enabled proper assembly of highly repetitive genomic regions such as the MHC [64], and existing tools for this purpose have not been applied extensively [65]. This deficit is particularly evident for bird species that have highly duplicated MHC genes. So far, *de novo* genome assembly in birds has predominantly relied on short-read sequencing technology and traditional assembly approaches, which results in the collapse of repeated sequences into a single location in the assembly. Bird MHCs are GC-rich, further hampering proper assembly [64] due to downward-biased sequence coverage.

Long-read sequencing technologies are quickly being adopted by the field of *de novo* genome assembly. Sequencing reads obtained through Pacific Biosciences Single Molecule Real-Time sequencing or Oxford Nanopore sequencing, for instance, now reach average lengths of several dozens of kilo base pairs or longer, dramatically improving the assembly of repetitive parts of the genome by anchoring repeats and duplicates to unique sequence content within the long reads [64]. In a number of primate species, long-read sequencing technology has already been employed to improve the characterization of complex gene families [66] including the MHC [67,68]. Similarly, in the newest chicken genome assembly, long-read sequencing has considerably improved the assembly of the MHC-Y region, extending it 1.5-fold in overall length [69]. Recent improvements in the accessibility of long-read sequencing technology are likely to result in the availability of many more high quality bird genomes, which offers promise for future comparative studies of MHC gene regions between species.

In addition to long-read sequencing, linked-read sequencing, such as offered by the 10x Genomics Chromium platform and other technologies in development [70], offer promising avenues for improving MHC assemblies. These technologies tag DNA molecules individually during library preparation and sequences them using short-read approaches in a cost-efficient manner, facilitating more efficient assembly and phasing, thereby reducing the assembly problem relative to phase-unaware approaches [71]. However, the best approach so far to document the catalog of MHC genes in any given species is likely to come from a combination of long-read sequencing and exhaustive short-read amplicon sequencing [72] (Westerdahl, unpublished). We expect such combined approaches and linked-read sequencing to complement and significantly assist the assembly of MHC regions based on long-read sequencing in the near future.

Nevertheless, avian MHC research has made use of short-read genome assemblies, which, at the same time, has revealed their limitations. A major PCR-screening of the avian tree of life for the presence of two ancestral MHC-IIB lineages was in part complemented by sequences retrieved from short-read bird genomes [34]. For several clades, the results from PCR screening of MHC-IIB genes were indeed confirmed in the genome assemblies, but only for species in which sequences of the two ancestral MHC-IIB lineages are highly divergent. In species with less divergent ancestral MHC-IIB lineages, the two regions were collapsed into a single assembled region (Burri, unpublished). Thus, genome assemblies based on short-reads can help confirm results obtained by other means, but the structure inferred from such genome assemblies may be biased toward models of multigene-family evolution that predict highly similar gene copies, such as concerted evolution, over others [73]. In birds, MHC-IIA genes are less well characterized than MHC-IIB, although initial evidence suggests that Pekin ducks (*Anas platyrhynchos*) and chickens only have a single MHC-IIA gene whereas the crested ibis has several sequentially repeated pairs of MHC-IIA and MHC-IIB genes [35,74,75]. Overall, it seems likely that MHC-IIA is less polymorphic than MHC-IIB in birds; however, further work is required to establish the general pattern for MHC-IIA genes across birds

### 2.2. Technical Advances: MHC Genotyping within Populations

As seen in Table 1, technological advances over the last two decades have brought about massive improvements in MHC genotyping methodology, resulting in an unforeseen resolution at the level of the individual [40,76,77]. In the 1990s, MHC genotyping was generally conducted with some form of fragment analysis, such as restriction fragment analyses (RFLP) and Southern blots using class I or IIB probes to visualize fragments [11,15,23]. However, with such approaches, a single allele could be represented by one or two bands, and the genetic polymorphisms indicated by the bands were not necessarily associated with coding DNA. PCR-based methods, whereby specific fragments of MHC genes are targeted and amplified, saw a focus on MHC-I exon 3 and MHC-IIB exon 2, which represent the most polymorphic exons of each class [27]. PCR-based amplicon sequencing, using degenerate or specific primers, has been the dominant method used to study MHC diversity in birds since. Meanwhile, methods used to characterize the PCR products have evolved, starting with cloning and Sanger sequencing, followed by various fragment conformation techniques (SSCP, DGGE, RSCA) [78,79,80] and, most recently, by high-throughput sequencing (Roche 454, Ion Torrent and Illumina) [81,82]. High-throughput amplicon sequencing catalyzed an enormous leap forward in terms of possible sequencing depth, enabling high-resolution MHC genotyping of individuals. This advance was especially advantageous for the songbird research community because many species within this clade have highly duplicated MHC genes.

The first generation of high-throughput MHC studies has revealed remarkably high levels of MHC diversity in many bird species [40,87,88,89,90,91,92], especially among passerine birds (Passeriformes). Despite early evidence from traditional MHC genotyping techniques that MHC diversity was higher in passerine than in non-passerine clades [24,93,94,95,96,97], the extreme levels of MHC diversity detected with high-throughput sequencing in some passerines was unprecedented and unexpected. For example, recent studies have found evidence for at least 33 MHC-I genes in sedge warblers (*Acrocephalus schoenobaenus*) [40] and 23 MHC-IIB genes in common yellowthroats (*Geothlypis trichas*) [76], numbers that have not been detected with traditional approaches (although these numbers may include both classical and non-classical MHC genes). Such numbers provide clear evidence that passerine MHCs are more gene-rich than those of galliforms, including the chicken [5,15,45]. Interestingly, thorough characterizations of the MHC genomic regions in some mammals and fish also report highly duplicated MHC-I and MHC-IIB regions [50,51].

An additional consideration for MHC genotyping using high-throughput amplicon sequencing is the bioinformatic processing of the sequence data. High-throughput sequencing produces vast amounts of data, but this comes at the cost of a relatively high error-rate [98,99]. Thus, it is necessary to remove artifactual reads via filtering to accurately estimate the number of MHC alleles detected. Although there is substantial variation in filtering approaches [100], a recent comparison of some of the most common filtering methods on the same sedge warbler dataset suggests that they yield similar estimates of the number of alleles per individual retained in the final dataset [101]. It is likely that the errors that are hardest to resolve for a typical MHC genotyping dataset are those arising during the PCR process [100,101]. Artifactual variants arising during the early stages of PCR may be highly represented in the final dataset and can appear very similar to true alleles, making them difficult to remove through filtering. Genotyping more than one preparation from the same individual can help identify PCR-generated errors [102]. Primer design can also have a dramatic effect on the accuracy of MHC genotyping. Unbalanced amplification efficiency across MHC alleles can lead to over-amplification of certain groups of genes and alleles while others may be missed—so-called ‘allelic dropout’ [30,102]. Thus, the use of multiple primer sets and the running of all samples in duplicates is advisable to gain the most complete and accurate survey of MHC alleles within and between individuals [30,88].

Since the arrival of high-throughput amplicon sequencing, there has been a huge increase in the number of bird species genotyped for MHC-I and -IIB genes, as seen in Figure 1: currently at more than 78 species for MHC-I and over 220 species for MHC-IIB [103]. This wealth of data has enabled comparative studies seeking to understand the extraordinary variation in MHC diversity we see across bird species [88,93,103]. When employing a purely bioinformatic approach to compile datasets for these analyses, such as harvesting the data from GenBank, it is important to exercise caution over the comparability of such data across species. As discussed above, there are many factors influencing the number of alleles detected in a typical MHC genotyping study, making it challenging to make between-species comparisons of MHC diversity across different studies using different amplification primers and methods [88]. However, in the largest study of this kind to-date, Minias and colleagues [103] implemented statistical approaches to account for variation in MHC diversity estimates generated by methodological differences between studies, demonstrating that it may be possible to remove some of the noise in comparative approaches. A wet-lab approach, whereby the MHC diversity estimate are generated within the same lab using comparably-designed primers and employing the same filtering steps, may result in more robust comparisons, but has the drawback of a more limited sample size [88,104].

## 3. Avian MHC Spreads Its Wings

### 3.1. High-throughput Studies of Avian Ecology and MHC

The ease with which individuals can be MHC-genotyped using high-throughput amplicon sequencing facilitates the search for associations between MHC variation and ecological traits in wild birds. The central role played by MHC molecules in adaptive immunity means that much of this work has focused on linking MHC variation or specific MHC haplotypes with resistance to pathogens [87,105,106]. One of the most well-studied pathogens in wild birds over the last two decades is avian malaria; the human studies reporting associations between specific MHC haplotypes and resistance to malaria have inspired such work in birds [21,107]. Qualitative resistance, in which the MHC haplotype is associated with the absence or clearance of malaria, and quantitative resistance, in which the MHC haplotype is associated with lower infection intensity of avian malaria parasites in the blood, have been studied in a wide range of passerine species, as seen in Table 2. There is not only evidence for particular MHC alleles or overall MHC diversity conferring qualitative and quantitative resistance to malaria but also evidence for potentially antagonistic effects, whereby certain alleles may confer resistance to one malaria strain but susceptibility to another strain [84,86,87,105,106,108,109,110,111,112,113]. A challenge with studies of avian malaria in the wild is that it is often difficult to distinguish between birds that have cleared the parasite after it has risen to high levels within individuals and birds that have resisted any accumulation of the parasite. Such uncertainties could be overcome by examining additional features of the immune response, such as testing birds for antibodies to malaria [114], which may be less likely to be detected if the pathogen never rose to high levels.

The above-mentioned associations between MHC and malaria resistance are correlational because the studies have been conducted on natural populations. Although there are significant advantages to studying wild animals subject to natural selection, this approach has its limitations, because it is difficult to be certain that MHC genes have a direct relationship to the pathogen/disease in question. However, in the case of Marek’s disease in the chicken, which is caused by an oncogenic herpesvirus, more detailed knowledge is available. Individuals with low cell-surface expression of generalist MHC molecules do not get sick (MHC-I haplotypes B21, B2, B6 and B14), whereas individuals with high cell-surface expression of specialist MHC molecules (B4, B12 and B15) often die from the disease [117,118]. Moreover, for mycoplasmal conjunctivitis in house finches, individuals with high MHC-IIB diversity have been experimentally shown to be more resistant to *Mycoplasma gallisepticum*, the bacteria responsible for this infection [119]. This study also found evidence for greater MHC-IIB diversity in wild house finch populations exposed to this disease when compared with naïve populations, further supporting the important role played by MHC [119]. However, resistance to *Mycoplasma* in house finches, paradoxically, induces reduced expression of MHCIIB genes and is governed by additional immune genes, including genes of the innate immune system [120,121].

### 3.2. Fitness Assocations with MHC from Ecological Studies

Beyond MHC associations with specific traits, the aim of most ecological studies of avian MHC is ultimately to understand the relationship between MHC variation and fitness. This question is best addressed using data from longitudinal studies that follow a population of wild birds over many generations. This type of data is extremely labor-intensive and time-consuming to collect, but there are a number of such projects which have produced valuable insights. Von Schantz was a pioneer in this field and reported that ring-necked pheasants with specific MHC haplotypes had longer spurs, were more attractive as mates, and had increased survival rates [13,14]. Higher survival and lifetime reproductive success have been linked to particular MHC variants in great tits [91], whereas no such patterns were seen in a natural population of collared flycatchers [109]. A study of yellowthroats (*Geothlypis trichas*) found that males with higher MHC-IIB diversity had greater survival in the wild [111]. A long-term study of great reed warblers has demonstrated possible sex differences in the fitness effects of MHC-I diversity: offspring recruitment success was higher in males with higher MHC-I diversity, whereas females with lower MHC diversity had higher offspring recruitment success [90]. This last study raises the intriguing possibility that there are sexually antagonistic effects of selection on MHC diversity.

It is possible that MHC-based mate choice plays an important role in maintaining the high degree of MHC diversity seen in wild populations [122,123], although results vary among studies. MHC-based mate choice can maintain MHC genetic diversity, not only by the selective advantage of specific MHC haplotypes but also through mate choice for partners with more diverse MHC genes or partners with MHC genes that are compatible with the chooser’s set of MHC genes, a type of disassortative mating leading to offspring with more divergent MHC genes [123,124]. Several studies have found evidence for each form of MHC-based mate choice in wild birds [85,125,126,127,128,129]. However, a recent meta-analysis investigating mate-choice driven by either MHC diversity or dissimilarity concluded that there is little evidence overall to support such mate-choice in birds [130]. MHC-based mate choice is still an active and ongoing field of research: 129 articles were published that matched the search terms “MHC”, “birds”, and “mate choice” in 2018 alone. Thus, it is likely that a clearer picture of the relationship between MHC and mate choice in birds will emerge in the future.

The high MHC polymorphism and the associations between MHC genes and fitness measures have led to interest in these genes from a conservation perspective, and it has been suggested that measures to maintain or promote MHC diversity should play a role in conservation efforts in endangered species [131,132,133,134,135]. Birds are no exception, with a number of studies on endangered or near threatened species, such as Hawaiian honeycreepers (*Drepanidinae*), New Zealand black robins (*Petroica traversi*), Seychelles warblers (*Acrocephalus sechellensis*), and crested ibis, focusing on MHC gene diversity [89,95,136,137].

### 3.3. Next Steps in Ecological Studies of MHC Variation

The ease of MHC genotyping using high-throughput amplicon sequencing is unfortunately not matched by our knowledge of the genomic structure of the MHC region in most birds or functionally important details such as expression differences between genes or even how sequence variation translates to differences in the antigen binding properties of MHC molecules. This knowledge deficit hampers our ability to interpret the biological relevance of the patterns revealed by MHC genotyping studies on most bird species.

A major restriction to estimating MHC diversity using high-throughput amplicon sequencing is an inability to assign alleles to specific genes in species with highly duplicated MHC genes. In these cases, differences in the number of MHC alleles between genotyped individuals will partly reflect heterozygosity or copy number variation. This ambiguity undermines the accuracy of counting the number of MHC alleles as an estimate of MHC diversity. Long-read sequencing technology will facilitate characterization of the MHC region across a wider range of bird species. A better understanding of the structure of MHC genes, haplotypes, linkage relationships, and the extent of recombination and interlocus gene conversion will lead to greater detail in studies of MHC correlates with fitness and disease in birds. In humans, different MHC genes (HLA-A, -B, and -C) encode different sets of MHC molecules, and genomic regions containing classical MHC genes with known immune function are more likely to be associated with disease resistance than regions containing non-classical MHC genes, which play a less well-characterized role in the immune system [56,138,139].

Ecological studies of avian MHC diversity are also hampered by a lack of knowledge of whether all MHC alleles are equally expressed. This is pertinent, because expression profiles relate directly to the function of MHC molecules. For example, classical MHC genes can have high or low expression, whereas non-classical MHC genes are defined by having generally low levels of expression [57]. Some basic information on expression, measured as transcription at the level of RNA, of MHC genes in birds beyond chickens is available [11,23,24,29,46,120,140,141,142], but less than a handful of studies have used high-throughput amplicon sequencing to address this question, and few studies have used specific organs, such as spleen [142,143,144,145], to measure expression, likely a more reliable guide to expression than whole blood, although further study is needed. Initial evidence suggests that, even in the case of passerines with numerous MHC genes, a high proportion of MHC-I gene copies are expressed [40,42,43,146], implying that the number of expressed MHC genes correlate with the number of MHC gene copies in the genome. The number of expressed MHC-I gene copies are thus likely to be higher in passerines than in galliforms. However, the degree of expression among passerine MHC genes may still vary. Although there is some evidence from house sparrows and tree sparrows (*Passer montanus*) of a similar single-locus dominance as chickens [43], Eurasian siskins (*Spinus spinus*) do not have a single dominantly expressed class I locus [42]. Thus, the question of whether single-locus dominance in MHC gene expression is generally the case across birds has yet to be conclusively answered. Differences in MHC gene expression may also occur depending on the sample used (blood versus various tissues, especially spleen) as well as the infection status of the individual sampled, although some studies have reported highly similar MHC-I expression profiles across tissues [35,41,42,120,147]. Further work is required to better understand how MHC gene expression differs across bird species and across organs. Adding MHC gene expression would add a novel dimension to the observed relationships between ecological traits and MHC diversity at the genomic level.

Finally, knowledge of how sequence differences between MHC genes actually affect the peptide-binding repertoire of MHC molecules is an overlooked area of research in most birds beyond the chicken. The existence of a crystal structure for chicken MHC-I molecules enables accurate predictions of the functional impact of MHC sequence variation in chickens because it demonstrates which amino acids are in close contact with the antigen and thus where MHC-I variation is likely to affect the peptide-binding repertoire [148]. This information led to the discovery of two distinct categories of MHC-I molecules in the chicken: those with promiscuous binding properties that are capable of binding a wide-range of antigenic peptides and those with fastidious binding properties that bind a narrow range of antigenic peptides [117]. In other bird species, in which there is limited knowledge of the structure of the MHC molecule and for which the precise peptide biding sites are unknown, it is more challenging to interpret the biological relevance of variation in MHC sequences. *In silico* prediction, whereby computer simulations attempt to model how sequence variation is likely to alter the peptide binding repertoire, opens up opportunities in this area [149] as it has in humans [150,151]. However, knowledge of the crystal structure of MHC in a wider variety of bird species would greatly improve the accuracy of such approaches.

## 4. Macroevolution of the MHC Across the Avian Tree of Life

### 4.1. Variation in the Number of MHC Genes Across the Avian Tree of Life

The study of MHC evolution across the avian tree of life provides promising insights into macroevolutionary links between genome structure, long-term forces molding gene number, and variation in life history traits. The evolutionary dynamics of the MHC are extraordinary, involving a combination of gene duplication, gene loss, gene conversion and different forms of recombination [152]. These mechanisms contribute to rapid expansions and contractions of MHC gene number and to high rates of sequence exchange within and between MHC duplicates, creating substantial variation in MHC sequence and haplotype structure [153,154,155,156]. As we have seen, variation in numbers of MHC genes across the avian tree of life is substantial [40,77,88,157,158], although still poorly known and likely to suffer from some degree of inaccuracy in birds outside of galliforms [30] with less well-characterized MHC genes.

Work spurred by the discovery of orthologous MHC-IIB genes across owls [159] led to the unveiling of two ancient MHC-IIB lineages in birds that originated before the radiation of extant birds [34,160]. This discovery was a major improvement in our understanding of MHC-IIB evolution across birds and has allowed researchers to ascribe new MHC-IIB genes to one of these two deep lineages in cases in which the sequence signatures that distinguish them have escaped concerted evolution [34]. Balasubramaniam et al. [104] found complex repertoires of MHC-IIB genes and high polymorphism in basal oscine songbirds of the largely Australian Corvides clade, suggesting that complex MHC structures are ancestral in oscine songbirds. A major gap in our knowledge of MHC evolution in passerine birds is in the suboscines [93], a diverse and largely tropical clade that deserves further study with regard to the immune system in general.

High-throughput sequencing approaches have accelerated surveys of MHC copy number across bird species, providing the first large-scale studies of MHC macroevolution. Studying 12 Passerida species, O’Connor et al. [88] suggested that MHC-I gene number was dominated by fluctuating selection and/or drift. However, in the most extensive survey of MHC gene copy number variation in birds thus far, Minias and colleagues [103] suggested that the evolutionary processes governing duplicate numbers differ between the two MHC classes. MHC-I gene number was found to be shaped by accelerated evolution and stabilizing selection (Figure 2, [103]), whereas MHC-IIB gene number appears to be shaped by fluctuating selection and genetic drift (Figure 3 [103]). The study by O’Connor et al. [88] generated MHC sequences in-house, using identical methods across species, thereby allowing more reliable comparisons across species; however, the larger sample size of Minias et al. (78 species) might have yielded a more robust result. Both studies confirm the extraordinary variation in MHC duplicate numbers across birds.

Many questions regarding the duplication history of the avian MHC remain open. First, the timing of duplications and the role of new duplication versus gene loss in governing MHC gene numbers are yet to be determined. Due to variable and often high rates of concerted evolution, sequence similarity alone provides no information on the age of MHC duplicates [161]. Highly similar duplicates may be recent and not diverged yet, or may be old, with gene conversion erasing sequence divergence in coding regions [30,31,34]. Only comparative genomic analyses based on comprehensive assemblies of the MHC region that include positional information on synteny, flanking sequences and orthology for each duplicate will provide the required resolution. Furthermore, confirmation of different modes of selection on MHC-I and MHC-IIB [103] will require additional data from less biased *de novo* genome-based approaches. With the data and sequencing methods used thus far, it is unclear whether evidence for different modes of selection on gene copy number between MHC-I and MHC-IIB could be driven by different rates of concerted evolution that determine the success with which MHC duplicates can be distinguished from one another.

### 4.2. MHC Gene Duplication and Life History

Some of the most pertinent questions regarding the evolution of the MHC multigene family concern the link between modes and histories of MHC evolution and the diversification of species and life histories. Have particular genomic structures of the MHC facilitated the exploitation of more diverse ecological niches and thus contributed to species radiations? For instance, might the high MHC gene numbers in passerine birds have played a role in the radiation of this most species-rich group of birds? Or enabled the evolution of life histories that risk exposure to larger or more diverse pathogen communities, such as encountered by species that perform seasonal migrations or breed in colonies?

A recent series of comparative analyses has investigated links between the MHC and life history evolution [103,162,163]. Minias et al. [162] suggested that evolution at pathogen-binding sites of MHC-IIB genes was faster and stronger in migratory and colonially breeding non-passerine birds in comparison with sedentary and solitary species. Additionally, using all available published data on avian MHC, Minias et al. [103] demonstrated that the number of MHC duplicates in 250 bird species correlated positively with lifespan and migratory behavior. O’Connor et al. [163] genotyped MHC-I loci in 39 species of Afro-Palearctic passerine birds and demonstrated that shifts in the number of MHC-I alleles are associated with the colonization of new environments as well the evolution of long-distance migration. Specifically, passerines that have escaped pathogen pressure, either by permanently colonizing the less pathogen-rich Palearctic from Africa, or by seasonally leaving Africa to breed in the Palearctic, show a reduction in number of MHC alleles over evolutionary time. The study by O’Connor et al. [163] also suggests that changes in life-history strategies in host species that affect the diversity of pathogens they encounter are strongly linked to MHC evolution. It is likely that, even with high rates of gene duplication, changes in MHC number probably do not evolve as fast as migratory behavior, which appears to change rapidly across the avian tree of life, sometimes over the span of a few decades [164,165]. However, several studies have reported substantial intra-species variation in the number of MHC alleles, and, by extension, copies per individual [40,86,90], raising the possibility that selection on standing genetic variation could potentially enable rapid adaptive shifts in MHC diversity. As with other areas of MHC evolution, more precise determination of MHC structure and gene number will help clarify relationships between MHC diversity and life history traits.

In summary, there is evidence for a relationship between MHC evolution and life history evolution in birds. Yet, the data that are available so far are restricted to a small number of studies which apply rudimentary approaches to studying MHC diversity. *De novo* genome sequencing using the latest long-read technology promises to provide crucial information on the actual number of MHC genes within genomes and, through information on gene orthology, insights into the timing of duplication. Data collected this way across the avian tree of life will provide unprecedented power to study macroevolutionary links between the MHC and life history evolution and speciation. Such studies might ultimately uncover links between MHC complexity and adaptive radiation across birds.

## 5. Conclusions

The avian MHC will continue to provide a major link between avian ecology and genetics, and harbors primary candidate genes for many behaviors of interest to ornithologists, including mate choice and disease resistance. The intersection between genetics and the ecology of birds is perhaps best captured by studies on the MHC. At the same time, the number of researchers pushing the boundaries to improve our understanding of avian MHC structure, expression, and evolution remains small. Although high-throughput PCR and sequencing surveys of MHC diversity have increased dramatically in recent years, such studies are unlikely to improve our understanding of the genomic structure of the MHC, and correlations with life history, mate choice, and disease resistance will remain tentative and uncertain due to our poor understanding of MHC gene number, linkage relationships, and other details.

We hope this update on MHC evolution in birds will spur further investigations of MHC structure, expression, function, and evolution in birds. What is sorely needed are robust genetic and physical maps of the MHC region in a handful of exemplar lineages of birds to provide anchor points for MHC studies in diverse species. Careful re-analysis and annotation of the wealth of short-read data, combined with comparisons between species, might alone yield substantial improvements in our understanding of MHC structure and evolution. Personal observations suggest that a substantial amount of gene structure data, including synteny information, can be gleaned from traditional short-read assemblies of bird genomes, particularly those with high coverage [166]. But, as this review has addressed, long-read data will accelerate and greatly improve our understanding of avian MHC evolution. Working both phylogenetically across species, as well as in-depth within key species with long-term ecological data sets, will undoubtedly provide new insights into the dimensions and consequences of MHC evolution in birds.

## Figures and Tables

**Figure 1 cells-08-01152-f001:**
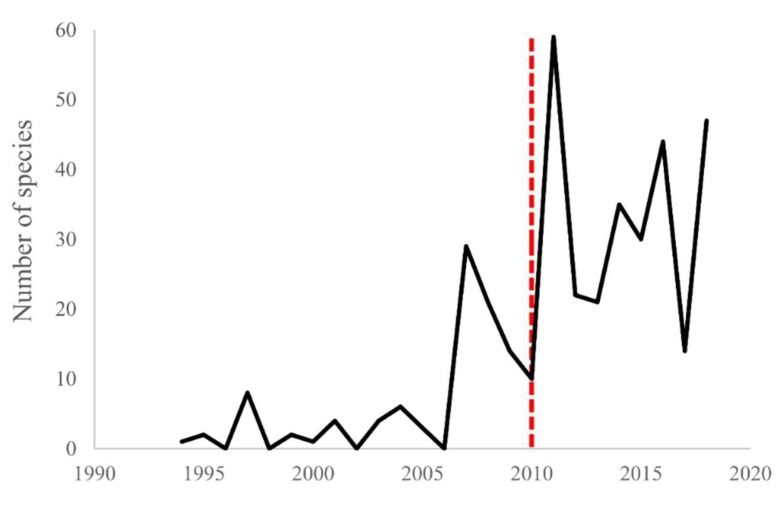
Evidence for the impact of high-throughput sequencing on the number of bird species genotyped for MHC, as demonstrated by the increase in the number of bird species genotyped for MHC each year since 2010 (red dashed line) when the first studies using high-throughput sequencing for avian MHC genotyping were published. Figure prepared using data from Minias et al. 2018 [103] with updated information added for studies until the end of 2018 using identical methods to those described within Minias et al 2018 [103].

**Figure 2 cells-08-01152-f002:**
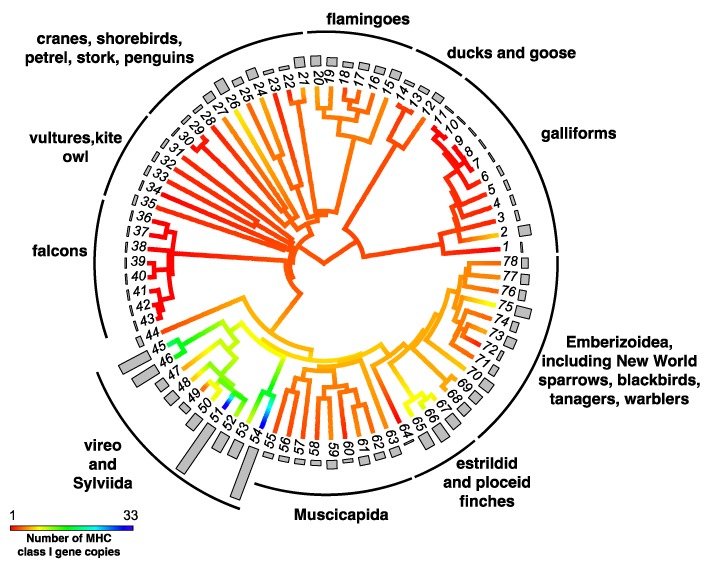
Ancestral character estimation of gene copy number at MHC class I genes along the branches and nodes of a tree for birds. Bars associated with each terminal node indicate the estimated number of MHC gene copies. Figure modified from Minias et al. 2018 [103]. Key to species names is supplied in Appendix A.

**Figure 3 cells-08-01152-f003:**
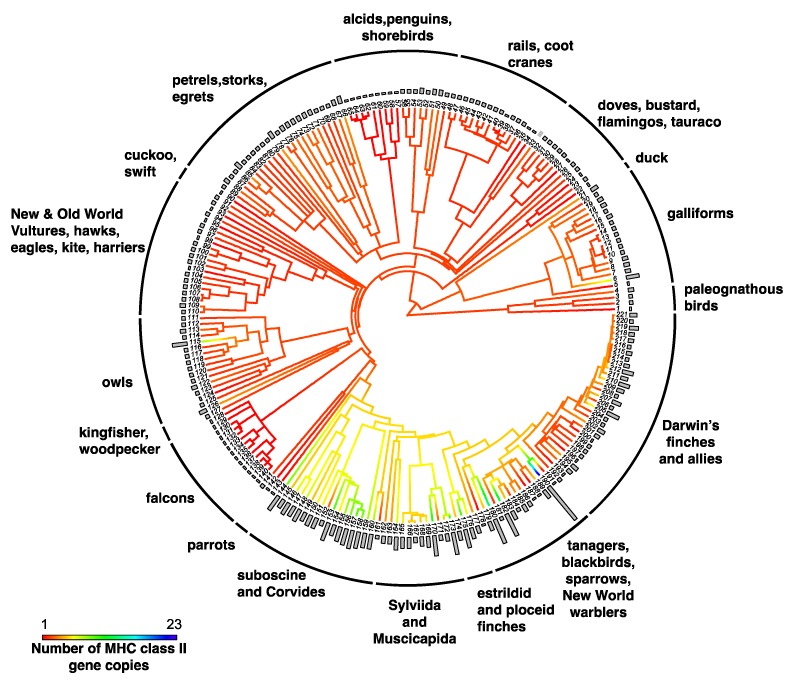
Ancestral character estimation of gene copy number at MHC class IIB genes along the branches and nodes of a tree for birds. Bars associated with each terminal node indicate the estimated number of MHC gene copies. Figure modified from Minias et al. 2018 [103]. Key to species names is supplied in Appendix A.

**Table 1 cells-08-01152-t001:** The development of major histocompatibility complex (MHC) genotyping methods in birds over four decades, with the advantages (Pros) and disadvantages (Cons) associated with different methods and future perspectives.

	Past		Present	Near Future	
	(1990s)	(2000s)	(2010+)	(2019+)	(2019+)
**Method**	Fragment analysis of genomic DNA (RFLP & Southern blot) [15,23]	Fragment analyses of PCR products (e.g., DGGE) [83,84,85]	High-throughput sequencing of PCR products (e.g., Illumina amplicon sequencing) [40,86,87]	Long-read sequencing (PacBio & Oxford nanopore)	Amplification of specific gene copies (High-throughput sequencing & qPCR)
**Pros**	No PCR artifacts Robust gene copy numbers	Moderate resolution	High resolution High coverage	No PCR artifacts Very long reads (100,000 bp possible) Gene synteny	Expression data Gene-specific amplification
**Cons**	Low resolution No sequence data	Artifactual alleles possible No sequence data	Artifactual alleles possible Short reads (up to 300 bp)	Low coverage Sequencing errors	Limited data on MHC diversity

**Table 2 cells-08-01152-t002:** Results of studies that have investigated the association between MHC and avian malaria. Resistance may be qualitative, whereby the malaria infection is cleared (prevalence), or quantitative, whereby the infection intensity is suppressed (intensity). Associations have been found between malaria and either specific MHC alleles, groups of MHC alleles that share similar antigen binding properties (‘supertypes’) or overall MHC diversity.

Species	Country	Class	Resistance	MHC Association	Reference
*Cyanistes caeruleus*	Sweden	MHC-I	Intensity	Alleles	Westerdahl et al. 2013 [110]
*Cyanistes caeruleus*	Spain	MHC-I	Prevalence Intensity	Alleles	Rivero-de Aguilar et al. 2016 [115]
*Parus major*	UK	MHC-I	Prevalence Intensity	Supertypes	Sepil et al. 2013 [87]
*Passer domesticus*	France	MHC-I	Prevalence	Alleles	Bonneud et al. 2006 [116] Loiseau et al. 2008; 2010 [84,108]
*Acrocephalus arundinaceus*	Sweden	MHC-I	Prevalence Intensity	Diversity Alleles	Westerdahl et al. 2005 [106] Westerdahl et al. 2011 [112]
*Acrocephalus schoenobaenus*	Poland	MHC-I	Prevalence	Supertypes	Biedrzycka et al. 2018 [105]
*Ficedula albicollis*	Sweden	MHC-IIB	Prevalence	Diversity	Radwan et al. 2012 [109]
*Geothlypis trichas*	US	MHC-IIB	Prevalence	Alleles Diversity	Dunn et al. 2013 [111] Whittingham et al. 2018 [86]
*Melospiza melodia*	Canada	MHC-I	Prevalence	Diversity	Slade et al. 2016 [113]

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
