# Peer review of "Avian MHC Evolution in the Era of Genomics: Phase 1.0"

_cells, 2019, doi:10.3390/cells8101152_

Round 1
Reviewer 1 Report
This is a nice review of what we know about Avian MHC diversity, and the likely benefits that will emerge from long-read genomic studies. I think many persons will benefit from the review. However, the authors leave out some important topics in the field of MHC studies, namely sexual selection and kin recognition, as well as the conservation aspects related to ecology. They are not really discussed, even though they are part of "ecology and evolution" (l. 56). I do not have a problem with this, and I can see the rationale for restricting the review to MHC structure and evolution, but I think the authors should point out more clearly in the Introduction that they are not covering these other areas of ecology and evolution.
My remaining comments are mostly minor points about some of the wording and choice of citations.
Title: consider changing the title, since there is really little discussion of "consequences". It could even be: "Avian MHC evolution in the era of genomics". Nice and simple.
37-42. This a long sentence and hard to follow. I would break it up into at least two sentences. I would suggest a period where the colon is after "mammals" on line 40. Also, the next part of the sentence would read better as: "mating preferences as well as disease resistance were revealed for the first time outside ..."
Then delete the part "arrived like a moonshot to bird biologists" or incorporate into the sentence in a different way (or use in another sentence). It is hard to read.
By the way, I would beg to differ about the importance of the Potts et al. Nature paper to bird biologists per se.
For bird biologists, I think the review by Jerry Brown and A. Eklund (Am. Nat. 1994) and Brown's subsequent review in Behavioral Ecology (1997) might have been more influential. Although the Potts et al. paper has been cited more (614 times) there are fewer citing papers (21.7%, n=133) that also have the keyword "bird" in them. In contrast, the Brown and Eklund (1994) paper has been cited 276 times and 84 of those references (30%) have the keyword "bird". So, on a percentage basis, I think more "bird" biologists have responded to the Brown and Eklund paper. This is a minor point, though. I think it would be interesting from a historical point of view to trace the study of MHC in birds/ wild animals back to its origins through various sub-fields (immunity, cancer genetics, sexual selection, evolution, kin recognition etc). But that is the subject of another paper!
Brown, J.L. and Eklund, A. (1994). Kin recognition and the major histocompatibility complex - an integrative review. Amer Naturalist, 143, 435-461.
Brown, J.L. (1997). A theory of mate choice based on heterozygosity. Behavioral Ecology, 8, 60-65. Cited 276 times
63-64. "restricted the progress that can be made using such approaches". The authors should consider rewording this sentence. I think many advances have been made with short-read sequencing, so to say that it has "restricted progress" is a bit too critical, I think. The sentence also refers to "using such approaches", which does not really make sense. I think the authors mean "future" advances? Maybe just delete that phrase?
74-75. I like the 'odd duck" joke. Actually, the chicken is not even that representative of galliformes, if you look at the number and arrangement of exons (eg., see Fig. 2 in Eimes et al. 2012. Immunogenetics. 65: 133-144). Quail are really different!
By the way, some of the references here (l. 74) are not really about Galliformes (26, 30) so I would cite Shiina et al. (2004, J Immunol 172: 5376-86), Eimes et al. (2012), and Ye et al. (2012, PlosOne 7: e32154).
117-126. With respect to long-read sequencing, it might be good to cite some of the recent papers that demonstrate the benefits more thoroughly, particularly in some other taxa or complex gene families. eg,
Larsen, P.A., Heilman, A.M. and Yoder, A.D. (2014). The utility of PacBo circular consensus sequencing for characterizing complex gene families in non-model organisms. Bmc Genomics, 15, 720.
Westbrook, C.J., Karl, J.A., Wiseman, R.W., Mate, S., Koroleva, G., Garcia, K., Sanchez-Lockhart, M., O’connor, D.H. and Palacios, G. (2015). No assembly required: Full-length mhc class i allele discovery by pacbio circular consensus sequencing. Human Immunology, 76, 891-896.
Gordon et al. (2016). Long-read sequence assembly of the gorilla genome. Science, 352.
194. add "the" so it is: "remove some of the noise"
323. "likely biased in birds outside of galliforms". Is "biased" the correct word? Maybe "higher", but I am not sure "biased" is appropriate.
Reviewer 2 Report
I am sorry, but I cannot recommend the manuscript by O'Connor et al. for publication in Cells. There already exist a number of excellent reviews on bird MHC, and I can't see how this new study makes a meaningful contribution. As the authors righteously conclude, the big problem for bulk analysis of avian MHC is the poor quality of many genomic data. Furthermore, the authors seem to enthusiastically list all reported associations between avian MHC and phenotypes without critically evaluating the statistics of those studies; such has been done already, and is not the point of having an expert reviewer summarizing data since I can rapidly make such summaries myself by just collecting abstracts from PubMed.
Reviewer 3 Report
In this review, the authors described update studies of MHC evolution in birds, including recent advances in our understanding of structural evolution of the avian MHC, as well as new insights into avian ecology and evolution provided by the MHC. In general, the manuscript is well written with MHC association studies on avian malaria and ecological patterns in nature such as lifespan and migratory behavior. Therefore, it will be informative to the researchers working in the field of MHC evolution and association of MHC alleles or haplotypes and disease or ecological traits, not only avian MHC but also other animal MHC. However, to clarify relationships between MHC and diseases or life history, it would be necessary to determine avian MHC genomic structure and expression of the MHC genes.
It would be helpful for readers unfamiliar with avian MHC genes to detail briefly classification of MHC-I, MHC-II B and MHC-Y genes. For instance, following three points might be unclear for the readers.
1) Are MHC-I genes classical and/or non-classical class I genes?
2) The authors could explain whether only MHC-II B genes encoding class II beta chains have been analyzed as avian MHC class II genes so far or not.
3) The authors could explain whether MHC-Y genes have been also identified as non-classical class I genes in other birds except chicken or not.
In ‘2. Avian MHC enters the genomic era’, most of descriptions have been well known as characterization of technical methods for analyzing MHC genomic structure and genotyping not only for avian but also other species. Therefore, I would recommend that the descriptions would be focused attention on the specific points on avian MHC.
In Table 1, I’d like to suggest that the authors some representative references could be added in each MHC genotyping method in birds. The references in the table will be help to understand what kinds of avian MHC genes and bird species were analyzed using the each method.
A huge increase in the number of bird species genotyped for MHC-I and -II B genes is shown in Fig. 1. To obtain the impact of high-throughput sequencing on the number of MHC alleles, it would be better to add the number of MHC-I and MHC-II B alleles in Fig. 1.
Fig. 2A and 2B are probably too small and therefore hard to read the names of bird species. I would strongly recommend that the names with high copy number at MHC class I would be shown with larger letters.
Round 2
Reviewer 2 Report
I recommended rejection as I did not see any benefit of this review. It is neither a convenient summary (if such is possible for avian MHC) nor an in depth critical analysis. But I am overruled by the other reviewers and by the editors, and the article is not wrong, so I will accept the paper for publication. I like that the authors now added a critical reference on MHC-based mate selection.